# Strains on the human femur after revision total knee arthroplasty: An in vitro study using digital image correlation

Elisabeth M. Sporer[1,2]◉*, Christoph Schilling[1]◉, Robert J. Tait[3], Alexander Giurea[4], Thomas M. Grupp[1,5]

**1** Aesculap AG, Research & Development, Tuttlingen, Germany, **2** Medical Department, Ludwig Maximilians University Munich, Munich, Germany, **3** Orthopaedic Institute of Henderson, Henderson, Nevada, United States of America, **4** Department of Orthopaedic Surgery, Medical University of Vienna, Vienna, Austria, **5** Department of Orthopaedic and Trauma Surgery, Musculoskeletal University Center Munich (MUM), Ludwig Maximilians University Munich, Munich, Germany

◉ These authors contributed equally to this work.
\* elisabeth.sporer@gmx.de

**Data Availability Statement:** All relevant data are within the manuscript and its Supporting Information files.

## Abstract

Pain at the tip of the stem of a knee prosthesis (End-of-Stem Pain) is a common problem in revision total knee arthroplasty (TKA). It may be caused by a problematic interaction between stem and bone, but the exact biomechanical correlate is still unknown. On top of this, there is no biomechanical study investigating End-of-Stem Pain at the distal femur using human specimens. Aim of this study was to find out whether the implantation of a revision total knee implant leads to high femoral surface strains at the tip of the stem, which the authors expect to be the biomechanical correlate of End-of-Stem Pain. We implanted 16 rotating hinge knee implants into 16 fresh-frozen human femora using the hybrid fixation technique and comparing two reaming protocols. Afterwards, surface strains on these femora were measured under dynamic load in two different load scenarios (climbing stairs and chair rising) using digital image correlation (DIC) and fracture patterns after overcritical load were analysed. Peak surface strains were found at the tip of the stem in several measurements in both load scenarios. There were no significant differences between the two compared groups (different trial sizes) regarding surface strains and fracture patterns. We conclude that implantation of a long intramedullary stem in revision TKA can lead to high surface strains at the tip of the stem that may be the correlate of femoral End-of-Stem Pain. This finding might allow for a targeted development of future stem designs that can lead to lower surface strains and therefore might reduce End-of-Stem Pain. Digital Image Correlation proved valid for the measurement of surface strains and can be used in the future to test new stem designs in vitro.

## Introduction

Total knee arthroplasty (TKA) is amongst the most commonly performed surgical procedures and is of substantial clinical and social interest [1]. Countries with consecutive registers, such

**Funding:** The author(s) received no specific funding for this work.

**Competing interests:** I have read the journal's policy and the authors of this manuscript have the following competing interests: Three of the authors (EMS, CS, TMG) are employees of Aesculap AG, a manufacturer of orthopedic implants. RT and AG are paid consultants for Aesculap AG. AG is receiving royalties from Aesculap AG and is an unpaid consultant for DePuy Synthes. He is member of the Austrian Orthopaedic Society and of "AE – Arbeitsgemeinschaft Endoprothetik". RT is receiving royalties from Conformis and is a paid consultant for this company. RT has stock or stock options in OnPoint Surgical and receives support from Conformis and ZimmerBiomet as Principal Investigator. TMG is scientific member of the working group „Evaluations & Studies" of the German National Joint Registry „Endoprothesenregister Deutschland" (EPRD), Advisory Board Member of the EU Consortium SPINNER "Next generation of repair materials & techniques for spine surgery" and Chair of working group I "Introduction of Innovations" of the "European Federation of National Associations of Orthopaedics and Traumatology" (EFORT) "Implant & Patient Safety Initiative". This does not alter our adherence to PLOS ONE policies on sharing data and materials.

as the USA, Australia, the United Kingdom, Germany and Sweden, have reported rising numbers of primary and revision TKAs over many years and the trend has been expected to continue in the future [1–6].

In revision TKA, secure implant fixation plays a special role. Because of soft tissue destruction and a lack of bone density and quality, hinged implants with long diaphyseal stems are used in most cases [7–9]. In the past, hinged prostheses have been improved noticeable. Meanwhile, they show a very acceptable long-term survival and are also seen as viable option in primary TKA [10–12]. Because of the increase in revision TKAs and the broader use of rotating hinge implants with diaphyseal stems in primary TKA, End-of-Stem Pain is a concern. End-of-Stem Pain is localized pain at the tip of the stem of a prosthesis. Sah et al., 2011 and Peters et al., 2005 reported End-of-Stem Pain only in 0 – 2.3% after revision TKA. Other studies observed prevalences up to over 20% at the tibia and over 10% at the femur [13–16]. The pain is described as mostly activity related [13, 15]. Patients with End-of-Stem Pain are dissatisfied with the result of the operation and feel restricted in their daily activities [13, 14]. Unfortunately, there is no widely accepted treatment option. In cases with persisting pain, re-revision is sometimes unavoidable.

Until now, the biomechanical correlate of End-of-Stem Pain was still unclear which prevented specific improvements in stem design and implantation process. In the clinical studies that have been conducted yet, no significant correlation between stem diameter, stem length and percentual canal fill with End-of-Stem Pain was found [13, 15, 16]. In a study by Barrack et al., patients with solid cobalt-chrome stems had significantly more pain than patients with slotted titanium stems, which the authors contributed to the lower elastic modulus of titanium compared to cobalt-chrome, but the specific stem design also seemed to have an influence [15]. Completo et al. measured strains on nine synthetic tibiae after the implantation of a conventional stem and a stem with a stem tip with decreased elastic modulus. Significant differences were only present when a massive varus load was applied [17]. In another study, the authors created finite element models and calculated that the maximum strains and contact pressures in tibiae after revision TKA were present at the tip of the stem [18]. We assume that End-of-Stem Pain is caused by a problematic interaction between stem and cortex, which leads to high femoral surface strains at the tip of the stem. These peak strains may be transferred to the periosteum which–in contrast to the bone itself—is sensible to pain. The finding of high surface strains at the tip of the stem may build the basis for the development of future stem designs optimized regarding lower surface strains and therefore with the potential to show a lower frequency of End-of-Stem Pain.

The specific aims achieved in this study were (a) to measure surface strains on human femora after implantation of a stemmed knee implant in a realistic scenario using digital image correlation and (b) to analyse the fracturing patterns of the specimens after overcritical load, to detect characteristic failure modes.

## Materials and methods

We performed an in vitro study using 16 fresh-frozen human femora from 8 donors [19]. The specimens were provided by the Medical University of Vienna. Donors with preexisting knee implants, obvious bone diseases or diseases that can affect the bone structure were excluded. The data were treated anonymously and the study was approved by the University of Vienna ethics committee (project no. 2018/2019). A rotating hinge revision knee implant with a slightly tapered cobalt-chrome stem of 177 mm length with 10 fluted grooves (EnduRo modular rotating hinge knee system, Aesculap AG Tuttlingen, Germany) was implanted into the femora by a senior orthopaedic surgeon (RT) familiar with the EnduRo implant. The stem

**Table 1. Overview of the specimens (n = 16).**

| Specimen | Knee | Group | Femoral component | Diameter of Trial | Diameter of Stem |
|---|---|---|---|---|---|
| small_trial_1 | Left | small_trial | F3 | 13,5 mm | 14 mm |
| standard_1 | Right | Standard | F3 | 14 mm | 14 mm |
| standard_2 | Left | Standard | F2 | 12 mm | 12 mm |
| small_trial_2 | Right | small_trial | F2 | 12,5 mm | 13 mm |
| small_trial_3 | Left | small_trial | F3 | 15,5 mm | 16 mm |
| standard_3 | Right | Standard | F3 | 16 mm | 16 mm |
| small_trial_4 | Left | small_trial | F3 | 17,5 mm | 18 mm |
| standard_4 | Right | Standard | F3 | 17 mm | 17 mm |
| standard_5 | Left | Standard | F2 | 15 mm | 15 mm |
| small_trial_5 | Right | small_trial | F2 | 14,5 mm | 15 mm |
| standard_6 | Left | Standard | F3 | 16 mm | 16 mm |
| small_trial_6 | Right | small_trial | F3 | 15,5 mm | 16 mm |
| standard_7 | Left | Standard | F3 | 20 mm | 20 mm |
| small_trial_7 | Right | small_trial | F3 | 19,5 mm | 20 mm |
| small_trial_8 | Left | small_trial | F3 | 19,5 mm | 20 mm |
| standard_8 | Right | Standard | F3 | 20 mm | 20 mm |

diameter (12 – 20 mm), the size of the femoral component (F2 – F3) and the anterior-posterior shaft offset (± 2 mm) were selected individually for each specimen. We used the hybrid fixation technique with a metaphyseal cementation and a cementless stem.

Two groups were compared. In the standard group (n = 8), the implantation was performed according to the standard protocol with a standard sized trial (1 mm increments). The surgeon selected the size of the final stem by a hand ream "feel" as he advanced the reamer. In the small_trial group (n = 8), more trial sizes were offered (0.5 mm increments) to see if "tactile feedback" during the stem selection process could be improved to enable optimized selection of the final stem size. To minimize the influence of confounders, one femur of each donor was assigned to the standard group, while the contralateral femur was assigned to the small_trial group (**Table 1**).

After implantation, the femoral heads and proximal metaphyses were removed, and the distal femora were embedded with polyurethane casting resin in a pot 247,4 mm proximally from the distal femoral cut (**Fig 1**). The distance between the tip of the stem and embedding was chosen as high as possible (40 mm) to improve differentiation between effects at the tip of the stem and artefacts at the embedding level. The cut of the medullary canal was sealed with modelling clay to prevent penetration of the resin into the medullary canal.

The femora were fixed on the test machine (DYNA-MESS Stolberg, Germany). The tibio-femoral contact force was applied in a sinusoidal waveform with a frequency of 1 Hz via the tibial component and the polyethylene gliding surface with a force application of 50% medial and 50% lateral.

Two scenarios were simulated. Though the pain is described as activity related, there is no information in the literature about specific activities that provoke End-of-Stem Pain [13, 15]. Therefore, we have decided for two scenarios that represent different biomechanical situations.

## ChairRise90˚

In order to simulate high bending moments in the sagittal plane at the tip of the stem, the femora were loaded with a 90˚ flexion angle (**Fig 2**). This scenario represents rising from a chair.

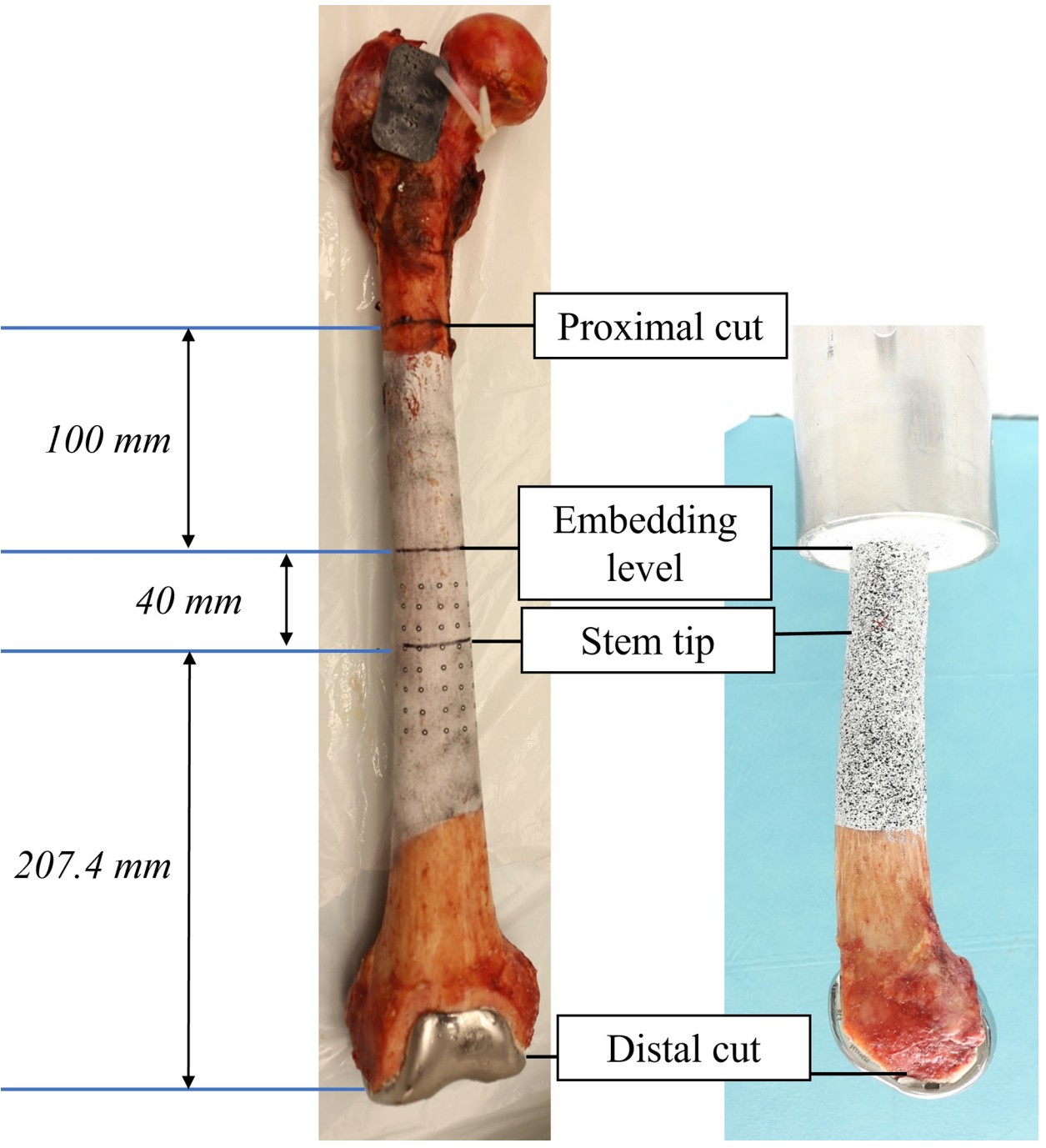

**Fig 1. Embedding process.** Embedding level: 247.4 mm proximally from the distal cut. Left: Anterior view, Right: Lateral view.

At this flexion angle, the axial force at the tibial side is 2061 N in a 75 kg person according to Bergmann et al. [20, 21]. It must be considered that the bending moments in the human femur are reduced substantially, especially by the patellofemoral counterbalance interaction. Therefore, we calculated a maximum force of 250 N to induce physiological bending moments at the tip of the stem. In the calculation, a patellar contact force of 2100 N at a 90˚ flexion angle, a patellar flexion angle of 70˚ and the femoral geometry of an exemplary specimen

(standard_2) were used [22–24]. The force was increased from 50 N to 250 N with steps of 50 N (100 sinusoidal load cycles per step).

### Stairs20˚

In order to simulate high axial forces at the tip of the stem the femora were fixed in a 20˚ flexed position. This represents climbing stairs [20]. The tibio-femoral load was increased from 1200 N with steps of 300 N (100 load cycles per step) until fracture of the bone. This allowed coverage of the whole physiological range and of different load situations in different patients (modes of failure).

The analysis of bone deformations during the scenarios described above was performed with digital image correlation (DIC). The measurement system was PONTOS 5 M (GOM Braunschweig, Germany) (Fig 2). The system recognizes points on the surface of the specimen by analysing the grey scale values of the pixels. By tracking these points during the load cycle with two cameras, information about 3-dimensional displacements and strains is received. To improve pattern recognition, after careful removal of soft tissue with a bone curette and cleaning the surface of the bone with isopropanol, a high-contrast black-and-white pattern was applied on the femora with aerosol spray before the experiments. A coordinate system was placed in the capture area (Fig 2). The y-axis was parallel to the axis of the stem and pointed towards distal, the x-axis pointed towards posterior and the z-axis towards medial or lateral, depending on the side of the leg. For data analysis the data were standardized to a right knee, so that the direction of the z-axis always pointed towards medial. The pictures were taken in a lateral view of the specimens. For the analysis, the 60[th] load cycle was tracked at each load level with a picture frequency of 15 Hz. From this picture series, the picture that was taken at the maximum of the sinusoidal load curve was chosen. This picture was used for the analysis of strains with Aramis Professional 2017 (GOM Braunschweig, Germany) by comparing it to a

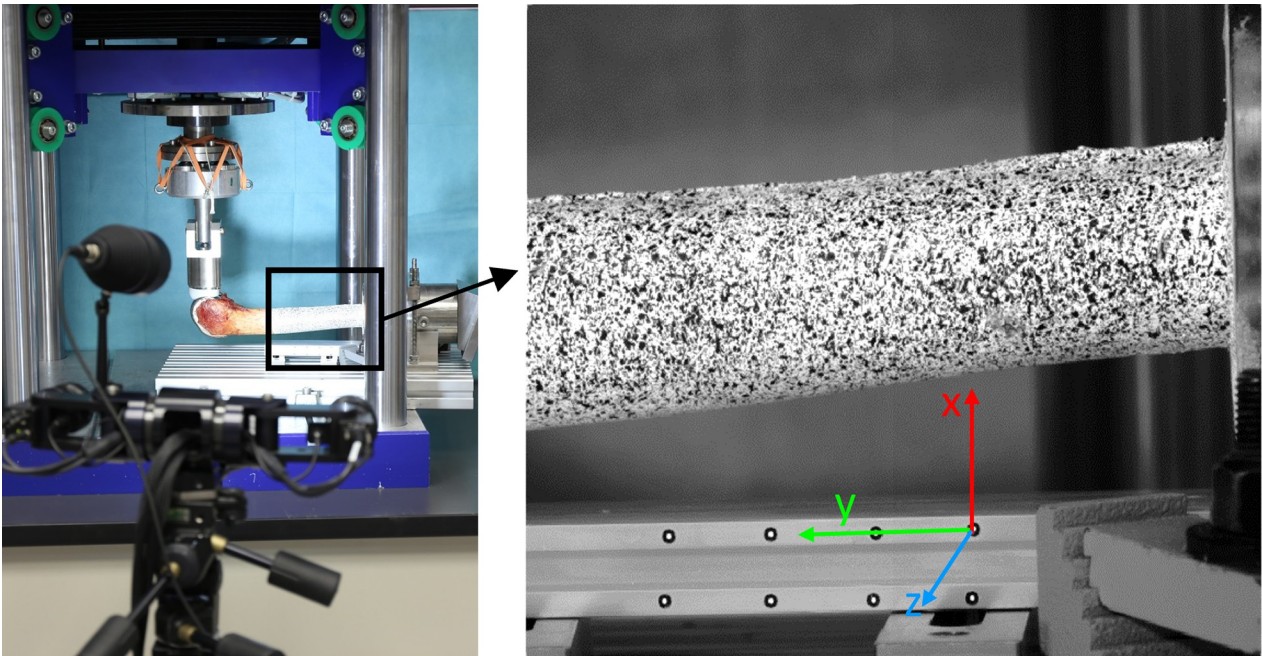

**Fig 2. ChairRise90˚.** Test-setup with dynamic compression-shear testing in a 90˚ flexed position for simulation of peak joint load when rising from a chair with load application 50% medial and 50% lateral and 3D deformation analysis system (left) using a high-contrast pattern (right).

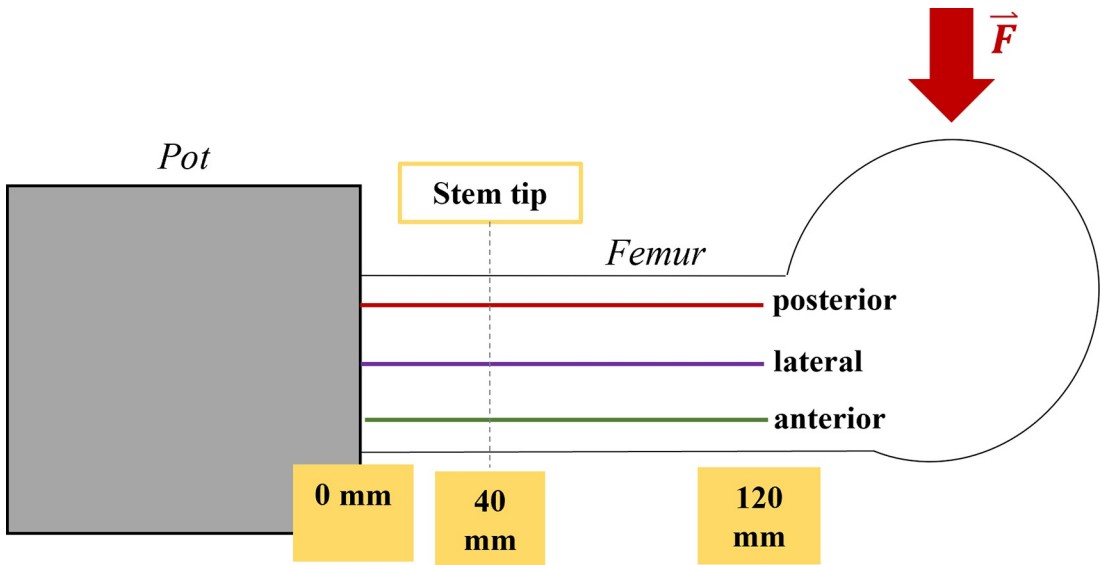

**Fig 3. Analysis of deformations.** The deformations are displayed along three sections (posterior, lateral und anterior). The axial height of 0 mm is at the embedding level, 40 mm at the tip of the stem, 120 mm is the distal end of the analysis. Red arrow: Application of the tibio-femoral contact force.

reference picture that was taken of each specimen before both load scenarios without preload. The reference length for strains was 1.74 ± 0.04 mm.

In the following, the displacements and strains are displayed along three longitudinal sections of the femur (posterolateral, lateral und anterolateral). For simplification, the posterolateral section is called posterior section and the anterolateral section is called anterior section. Along these sections, the axial height 0 mm is at the embedding level, 40 mm at the tip of the stem and 120 mm at the distal end of the analysis (**Fig 3**). Along these sections, the deformations were measured in steps of 0.1 mm. Outliers in the strain measurement with a deviation of more than 300 μm/m compared to the previous value were eliminated.

A moving average was calculated over 50 data points.

The statistical analysis to test for differences in strains between the standard group and the small_trial group at the tip of the stem (axial height 30 mm to 50 mm) was performed using Statistica 10 (StatSoft Europe GmbH Hamburg, Germany). An analysis of variance (ANOVA) was conducted with a significance level of $p < 0.05$. Before the analysis, the normal distribution of the data (normal p-p plots; $p < 0.05$) and the homogeneity of variance (Levene-test) were verified. After this, the Scheffe-Test was conducted as post-hoc-test.

After the Stairs20˚ test, the fracturing patterns of the bones were analysed.

## Results

### A: Surface strains under dynamic load

In both scenarios, the highest load level was used for the analysis (ChairRise90˚: 250 N, Stairs20˚: Load level before fracture). In the following, the strain in direction of the y-axis (length axis of the femur) is called longitudinal strain. The strain perpendicular to this strain is called radial strain.

**ChairRise90˚.** The longitudinal strains increased from distal towards the embedding level (**Fig 4, top**). Posterior, a longitudinal lengthening and a radial compression was measured. Anterior, a longitudinal compression and a radial lengthening was found (**Fig 4, top**). The

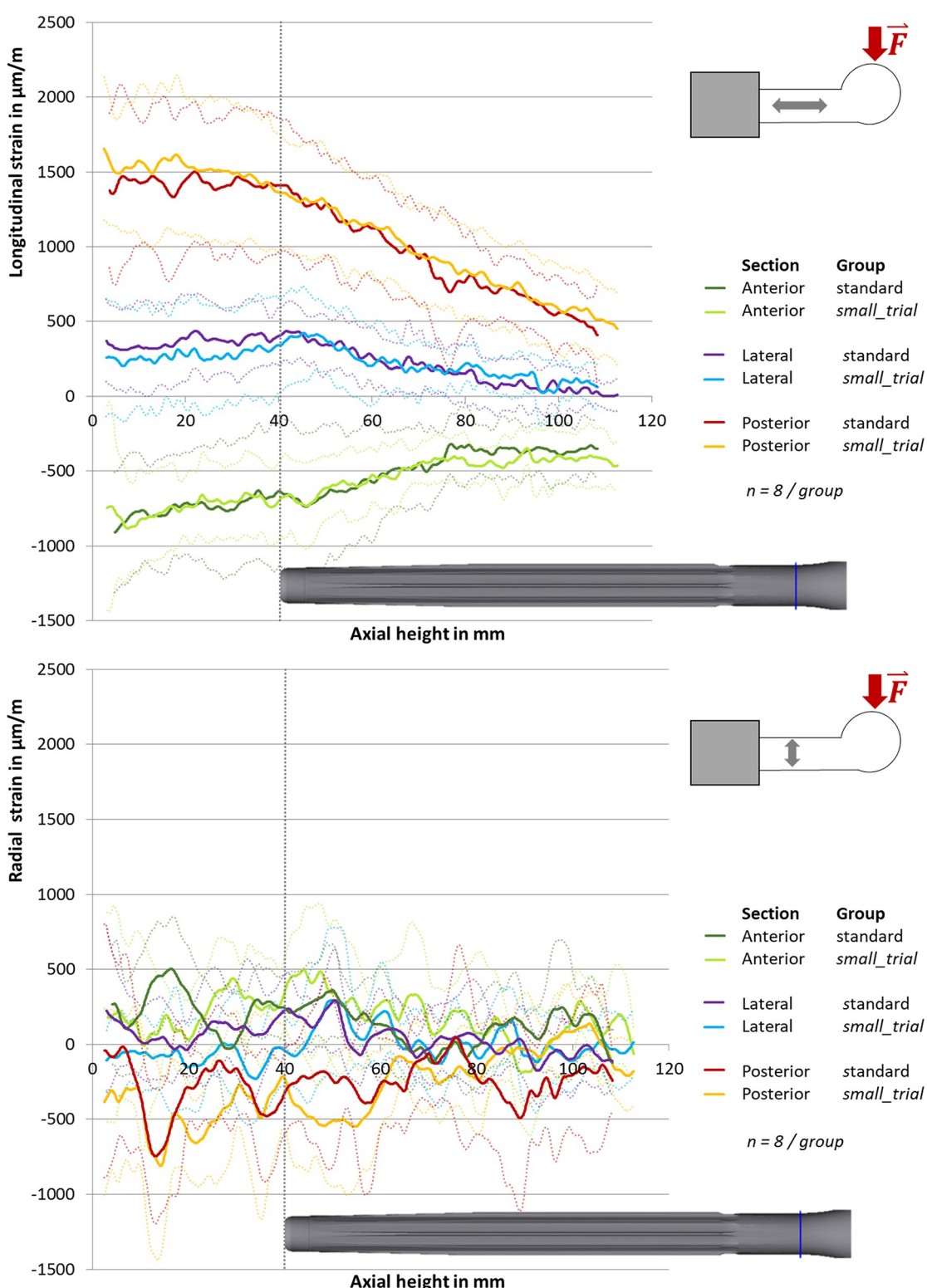

**Fig 4. Strains in ChairRise90˚.** Longitudinal strain (top) and radial strain (bottom) in µm/m with standard deviation (dotted line). 0 mm is the embedding level, the tip of the stem is at 40 mm.

maximum strain was a longitudinal strain of 1580 ± 450 μm/m (axial height 2.5 mm, n = 16) (**Fig 4, top**). Posterior, a change in the slope of the curve at the tip of the stem was registered. While the strains increased significantly from distally to the tip of the stem, they stayed on a stable level from the tip of the stem to the embedding level (**Fig 4, top**). Laterally, a maximum of strain was found at the tip of the stem in the longitudinal direction as well as radially. The average (n = 16) maximum lateral strain was 416 ± 250 μm/m in the longitudinal direction (axial height 44.3 mm) and 289 ± 424 μm/m radially (axial height 50.4 mm) (**Fig 4, top**). An increase in radial strains at the tip of the stem was also found anterior. The maximum strain there was 397 ± 379 μm/m (axial height 47.5 mm, n = 16) (**Fig 4, bottom**). Comparing the two implantation methods, no significant difference in strains was found between the standard group and the small_trial group at the tip of the stem (axial height 30 mm to 50 mm, p > 0.15).

**Stairs20˚.**   The strains increased from distally towards the embedding level. As in Chair-Rise90˚, a longitudinal lengthening and a radial compression was found posterior, as well as a longitudinal compression and radial lengthening anterior. (**Fig 5**). The maximum strain value of 7169 ± 2249 μm/m was measured posterior in the longitudinal direction close to the embedding level (axial height 3.3 mm, n = 16) (**Fig 5, top**). As seen in **Fig 5**, strain measurement in the longitudinal direction showed a change in slope of the lateral graph at the tip of the stem. The average longitudinal strain at the tip of the stem (axial height 40 mm, n = 16) measured 4910 ± 1672 μm/m posterior, 2427 ± 1261 μm/m lateral and -1483 ± 1470 μm/m anterior.

Radially, an anterior strain maximum of 779 ± 668 μm/m was present at the tip of the stem (axial height 39.4 mm, n = 16). As in ChairRise90˚, no significant differences between the standard group and the small_trial group were found (0.27 ≤ p ≤ 0.94).

## B: Fracturing patterns at supercritical load

In four specimens, the implantation caused a longitudinal fracture at the anterior side of the femur (**Fig 6**). The fractures measured 5 to 7 cm. The preexisting fractures did not lead to earlier failure in the load scenarios. Also, in Stairs20˚, the affected femora did not fracture along the preexisting fracture line. Two out of the four specimens with a longitudinal preexisting fracture were in the standard group, while two were in the small_trial group.

The fracturing load ranged from 1500 N to 4800 N (**Table 2**).

Both contralateral femora of the same donor fractured at similar loads. Several characteristic courses of the fracture lines were observed. The most frequent fracture pattern was characterized by a horizontal fracture of 20 – 30 mm length at the anterior tip of the stem. This pattern is called "stem pattern" (**Fig 7**). Other specimens showed wedge fractures with an anterior wedge and a posterior horizontal fracture line between the tip of the stem and the embedding (**Fig 8**). Four specimens fractured at the embedding level. Two specimens had a special pattern with several bone fragments or a strong torsion and could not be assigned to one of the other groups. **Table 2** lists the specimens with the corresponding fracture load, as well as the fracture pattern. In cases where the specimens fractured during the load increase between two load levels, both load levels (lower / upper level) are listed.

## Discussion

The overarching goal of this study was to find a biomechanical correlate of End-of-Stem Pain in the human femur and thus, build the basis for future improvements in stem design. Therefore, we (a) measured strains on 16 human femora under dynamic load after revision total knee arthroplasty in vitro and (b) analysed the fracture patterns of these femora after overcritical load to detect characteristic failure modes.

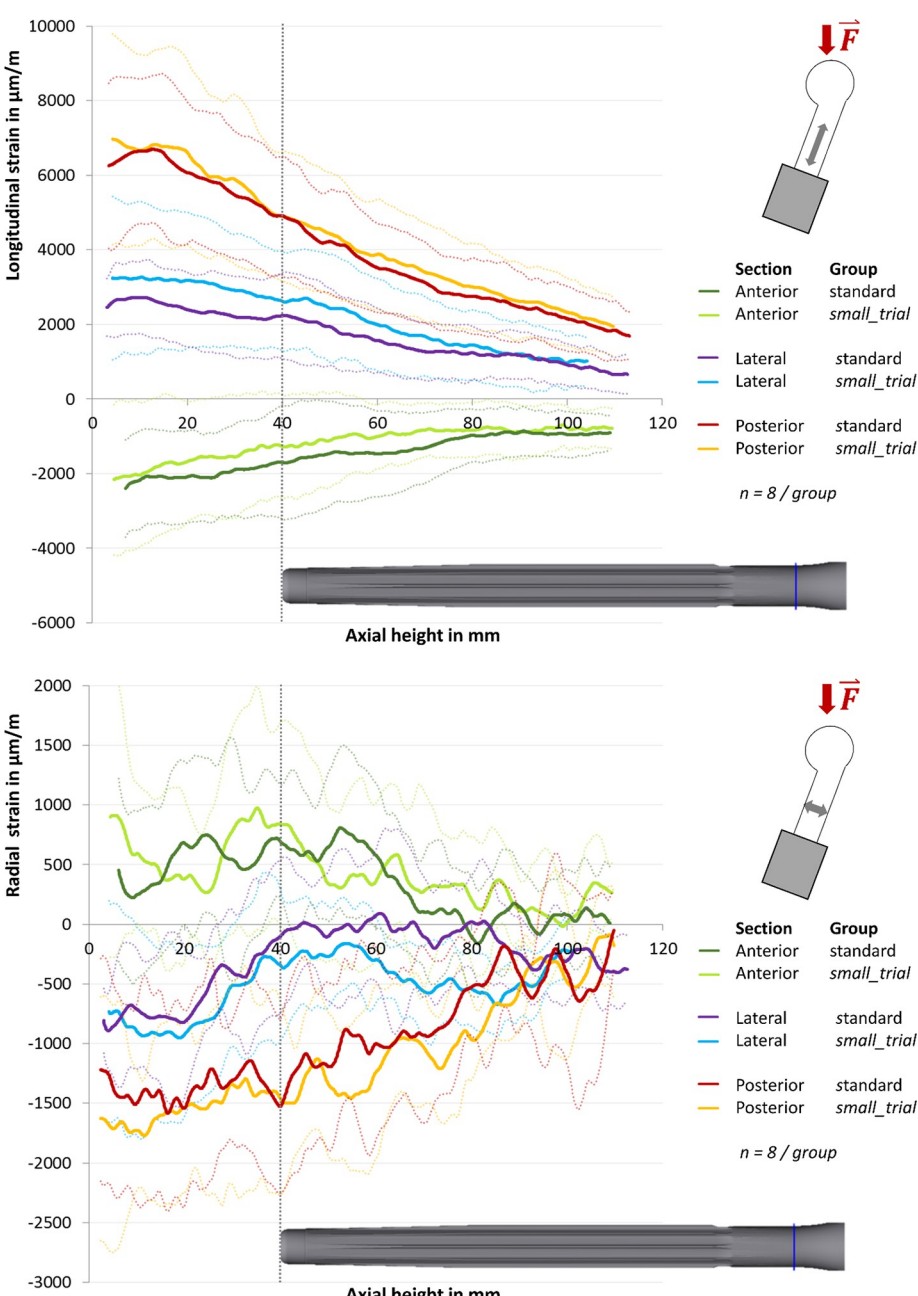

**Fig 5. Strains in Stairs20˚.** Longitudinal strain in μm/m with standard deviation (dotted line). 0 mm is the embedding level, the tip of the stem is at 40 mm.

Different causes of End-of-Stem Pain have been discussed in the literature [15, 17, 18]. The bone itself is not sensible to pain, in contrast to the thin layer of periosteum that covers the surface of the bone. Thus, it is striking that the underlying process of End-of-Stem Pain affects the surface of the bone and the periosteum. Our hypothesis is that the implantation of a long diaphyseal stem can lead to pathological surface strains, which can cause End-of-Stem Pain. As End-of-Stem Pain occurs predominantly during activity and seldom at rest, a major advantage of this study is that the measurement was performed under dynamic load [13, 15].

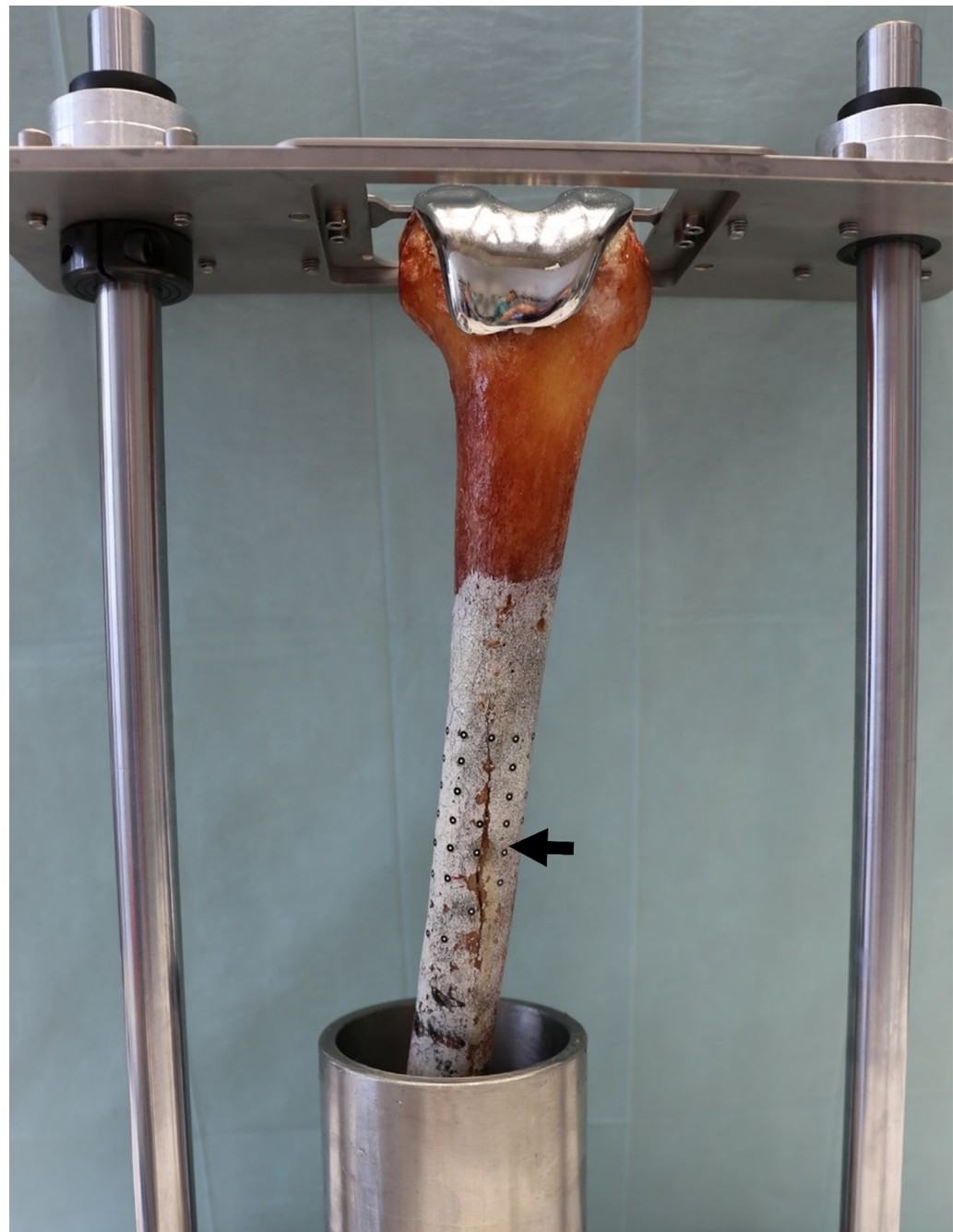

**Fig 6. Preexisting fracture.** Longitudinal fracture of 7 cm length at the anterior femur, caused by the implantation of the stem.

In the research of End-of-Stem Pain, our study is the first one to use digital image correlation (DIC) for strain measurement. Strain gauges have been considered the gold standard for strain measurement for a long time, but they only offer strain measurement at discrete points. DIC is a non-contact method to measure strains and it allows for a full-field strain analysis.

The method has been successfully used for strain measurement in earlier studies [25–31]. In a study by Correa et al. correlation between strain measurement using strain gauges and

**Table 2. Fracture forces and fracture patterns.**

| Specimen | Fracture force | Fracture pattern | Preexisting fracture from implantation |
|---|---|---|---|
| small_trial_1 | 3300 / 3600 N | special pattern | / |
| standard_1 | 4200 N | stem pattern | / |
| standard_2 | 1800 / 2100 N | embedding | / |
| small_trial_2 | 1800 N | stem pattern | / |
| small_trial_3 | 3000 N | stem pattern | / |
| standard_3 | 2700 N | embedding | / |
| small_trial_4 | 1800 / 2100 N | wedge fracture | Longitudinal fracture anterior (length: 6,5 cm) |
| standard_4 | 2100 / 2400 N | wedge fracture | / |
| standard_5 | 1500 / 1800 N | wedge fracture | Longitudinal fracture anterior (length: 7 cm) |
| small_trial_5 | 1800 N | stem pattern | / |
| standard_6 | 1800 / 2100 N | stem pattern | Longitudinal fracture anterior (length: 6 cm) |
| small_trial_6 | 2100 N | stem pattern | Longitudinal fracture anterior (length: 5 cm) |
| standard_7 | 4500 N | embedding | / |
| small_trial_7 | 3900 N | stem pattern | / |
| small_trial_8 | 4800 N | wedge fracture | / |
| standard_8 | 4800 / 5100 N | embedding, special pattern | / |

DIC has been shown [29]. With the system used in our study an approximated accuracy for strain of 0.1% was achieved for both in-plane and out-of-plane measurement. The reproducibility is approximately +/- 0.2%.

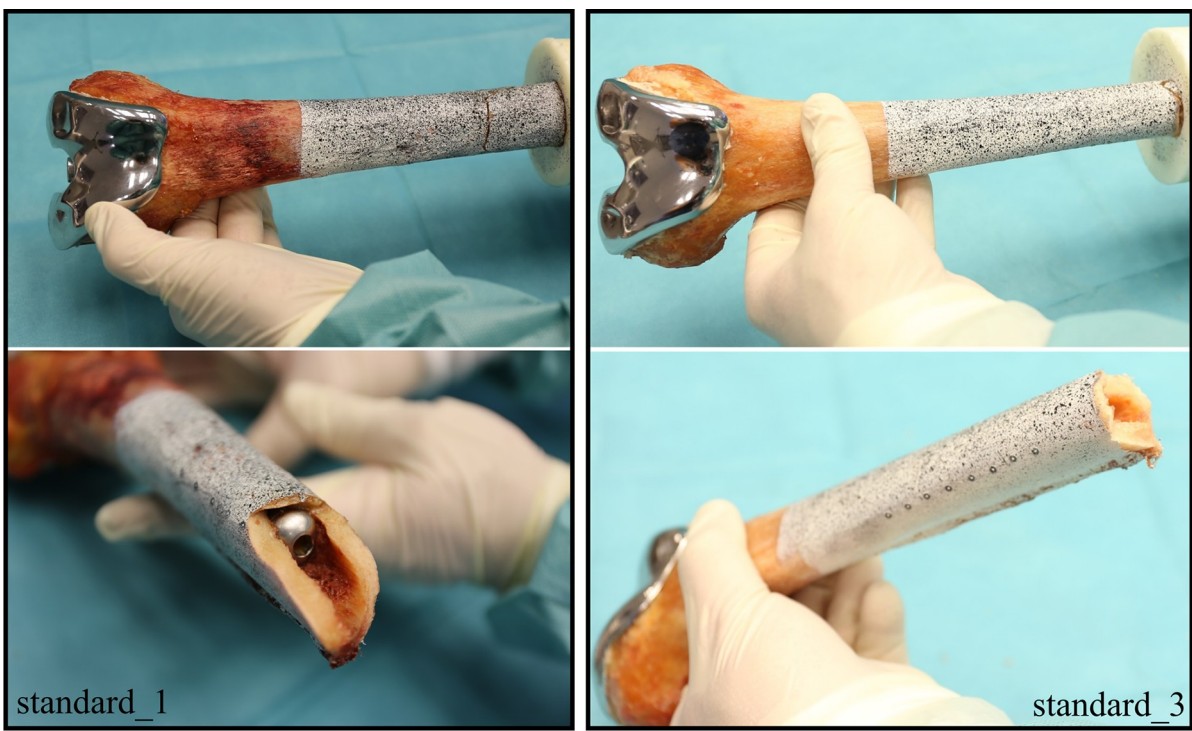

**Fig 7. Different fracture patterns.** Left: „Stem pattern"with anterior horizontal fracture line (standard_1). Right: Fracture at the embedding level (standard_3).

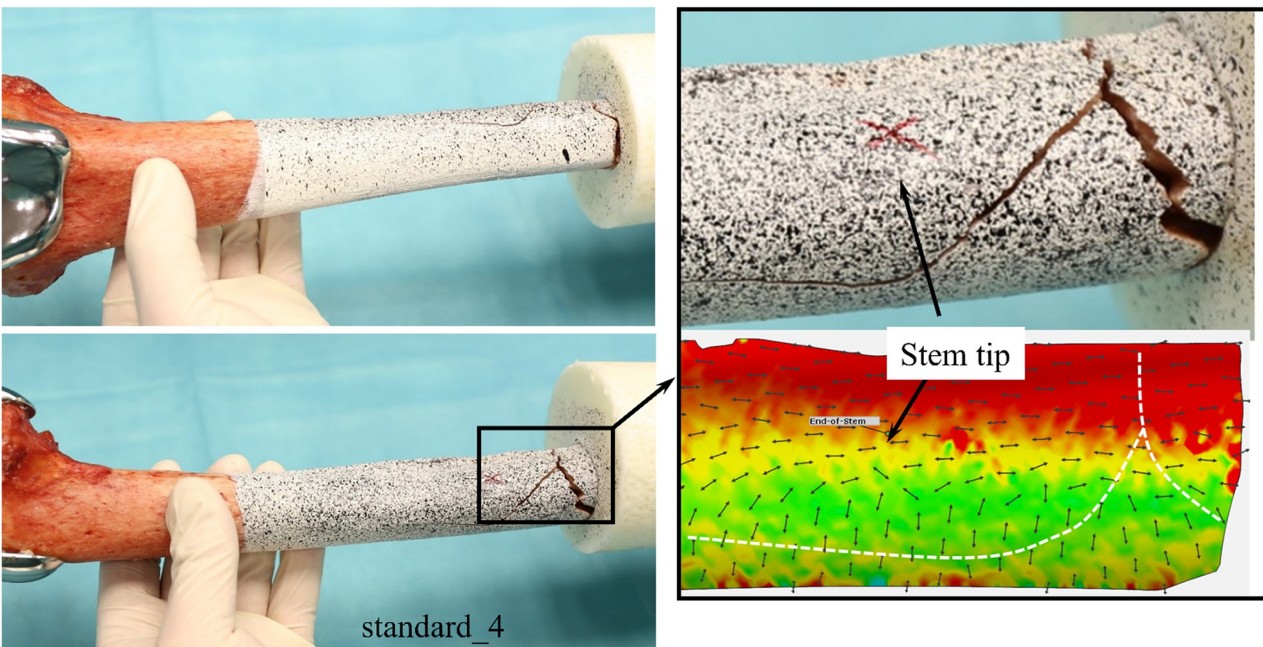

**Fig 8. Wedge fracture.** Left: Wedge fracture from anterior (upper picture) and from lateral (lower picture). Right: Detailed lateral view (upper picture) and analysis of strains in load level before fracture (lower picture): Small black arrows indicate major strains; red areas indicate high strains while green areas indicate low strains. Fracture line is represented by white dashed line (standard_4).

A limitation of our study is the lack of a comparison to the physiological situation. In following studies, this gap should be closed.

The surface strains increased from distally towards proximally. This was expected, as the lever arm of the force that was applicated distally increased towards the embedding level. Therefore, the bending moments and consequently the bending strains increased towards proximally. In ChairRise90˚ and Stairs20˚, the femora were strained in a longitudinal direction and compressed radially at the posterior section of the bone, while they were compressed longitudinally and strained radially at the anterior section of the bone. This is also in line with the expectation.

As stated in our hypothesis, there were characteristic strains at the tip of the stem. In ChairRise90˚, a positive maximum of longitudinal strains, as well as of radial strains, was found at the lateral tip of the stem. In both load scenarios, a radial maximum of strains was measured at the anterior tip of the stem.

The cause for these high strains at the tip of the stem remains subject to discussion. First finite element analysis (FEA) simulations, performed by the authors of this study, indicate that the tip of the stem moves away from its anterior contact point with the inner cortex during weight bearing. As the bend of the femur decreases during weight bearing, the stem is less clamped in the medullary canal and the anterior tip of the stem moves away from its anterior contact point. This might lead to a rolling mechanism of the stem at the anterior or posterior cortex and therefore might induce peaks in surface strains on the femur. No significant differences were found between the standard implantation and the small_trial group which indicates that 0.5 mm increments in trial size are not necessary and that the size of the trial has no effect on surface strains. The experimental finding of strain concentrations at the tip of the femoral stem is new and has important implications for future research.

The peaks in strains at the tip of the stem are in concordance with a study by Kim et al. [18]. In this FEA-study, Kim et al. calculated that the contact pressures and von Mises stresses were especially high at the tip of the stem. Comparing the results with a study performed by Completo et al., the measured strains reach a similar size [17]. While we measured strains in longitudinal and radial direction, Completo et al recorded principal strains. The maximum strain at the tip of the stem in our study was 1387 ± 423 μm/m posterior in ChairRise90˚ and 4910 ± 1672 μm/m posterior in Stairs20˚. The minimum strain was -667 ± 401μm/m anterior in ChairRise90˚ and -1483 ± 1470 μm/m anterior in Stairs20˚. The maximum and minimum principal strains in the study by Completo et al. were approximately 1600 μm/m lateral and -2000 μm/m medial when a varus load was applied and 450 μm/m and -1300 μm/m posterior when a normal physiological load was applied. It has to be considered that Completo et al. used tibiae and the specimens were made from synthetic material. Also, in Stairs20˚ the load level before fracture was analyzed, which explains why the observed strains were higher in our study. In Completo et al.´s study, surface strains were significantly higher in specimens after implantation of a prosthesis with a stem in comparison to specimens without a stem, but the differences were only present when a massive varus load was applied (load application 100% medial). In our study, we were able to show that high strains at the tip of the stem are not only present in synthetic tibiae but also in human femora. Our study does not investigate which level of strain results in pain. To our knowledge there are also no previous studies that report which level of strain results in pain, neither at the human femur nor at other bones or tissues. Further studies are required in this field.

In four specimens, we observed anterior longitudinal fractures, caused by the implantation of the stem. This illustrates the substantial radial forces that appear during the implantation process, which is in line with previous findings of substantial contact of the tip of the stem with the anterior cortex [19]. On the other hand, the preexisting fractures did not lead to earlier failure in the Stairs20˚ test. The preexisting fractures obviously do not decrease bone stability under these loading conditions. Nevertheless, the question arises which role the preexisting fractures play in the genesis of End-of-Stem Pain.

As in Stairs20˚, both autologous femora of one donor fractured at similar load levels, the effect of the implantation method (standard, small_trial) is inferior to the interindividual differences in femoral geometry. While we expected a direct fracture of the femora at the embedding level, different fracturing patterns were observed. It was striking that the fracture line of the "stem pattern" was located horizontally at the tip of the stem. Some of the wedge fractures also showed a change in direction of the fracture line at the tip of the stem. Until now, the amount of biomechanical influence of the stem in knee arthroplasty has still been unknown. The specific fracture pattern found in this study suggests that there is a high biomechanical impact of the stem. It is also still unclear why only certain patients suffer from End-of-Stem Pain whereas others do not experience End-of-Stem Pain. In our study, only in some specimens overcritical load led to the stem pattern fracture. This may suggest that the biomechanical influence of the stem is higher in some patients than in others. This may explain why only some patients suffer from End-of-Stem Pain and it underlines the importance of individualized selection of the right implant.

The observed results indicate that End-of-Stem Pain may be caused by peak surface strains of the bone at the tip of the stem. This is the basis for a novel approach in the development of future stem designs and a reduction of End-of-Stem Pain. In subsequent studies, a FEA-model will be developed based on CT-data of the specimens that were part of this study. The model can be used to simulate surface strains on the femur under load. This will allow digital testing of innovative stem materials, dimensions, and designs regarding the resulting surface strains which can be validated with the introduced in vitro technique.

## Supporting information

**S1 File. Strains in ChairRise90˚.** Radial and longitudinal strains at the highest load level in % for all specimens with mean values and standard deviation.
(XLSM)

**S2 File. Strains in Stairs20˚.** Radial and longitudinal strains at the highest load level in % for all specimens with mean values and standard deviation.
(XLSM)

## Author Contributions

**Conceptualization:** Christoph Schilling, Robert J. Tait, Alexander Giurea, Thomas M. Grupp.

**Formal analysis:** Elisabeth M. Sporer, Christoph Schilling.

**Investigation:** Elisabeth M. Sporer, Robert J. Tait.

**Methodology:** Elisabeth M. Sporer, Christoph Schilling, Thomas M. Grupp.

**Project administration:** Thomas M. Grupp.

**Resources:** Alexander Giurea, Thomas M. Grupp.

**Supervision:** Thomas M. Grupp.

**Visualization:** Elisabeth M. Sporer.

**Writing – original draft:** Elisabeth M. Sporer.

**Writing – review & editing:** Elisabeth M. Sporer, Christoph Schilling, Robert J. Tait, Alexander Giurea, Thomas M. Grupp.

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
