## [Decision Letter · Decision Letter 0]

18 Jan 2024

PONE-D-23-39275Strains on the human femur after revision total knee arthroplasty: An in vitro study using digital image correlationPLOS ONE

Dear Dr. Sporer,

Thank you for submitting your manuscript to PLOS ONE. After careful consideration, we feel that it has merit but does not fully meet PLOS ONE’s publication criteria as it currently stands. Therefore, we invite you to submit a revised version of the manuscript that addresses the points raised during the review process.

We look forward to receiving your revised manuscript.

Kind regards,

Pawel Klosowski, D.Sc.

Academic Editor

PLOS ONE

[I have read the journal's policy and the authors of this manuscript have the following competing interests:

Three of the authors (EMS, CS, TMG) are employees of Aesculap AG, a manufacturer of orthopedic implants.

RT and AG are paid consultants for Aesculap AG. AG is receiving royalties from 

Aesculap AG and is an unpaid consultant for DePuy Synthes. He is member of the Austrian Orthopaedic Society and of ”AE – Arbeitsgemeinschaft Endoprothetik”. RT is receiving royalties from Conformis and is a paid consultant for this company. RT has stock or stock options in OnPoint Surgical and receives support from Conformis and ZimmerBiomet as Principal Investigator. TMG is scientific member of the working group „Evaluations & Studies“ of the German National Joint Registry „Endoprothesenregister Deutschland“ (EPRD), Advisory Board Member of the EU Consortium SPINNER “Next generation of repair materials & techniques for spine surgery” and Chair of working group I “Introduction of Innovations” of the ”European Federation of National Associations of Orthopaedics and Traumatology” (EFORT) “Implant & Patient Safety Initiative”.]. 

5. Please include the reference section of your manuscript.

6. We note that Figure 1, 2, 6, 7, and 8 in your submission contain copyrighted images. All PLOS content is published under the Creative Commons Attribution License (CC BY 4.0), which means that the manuscript, images, and Supporting Information files will be freely available online, and any third party is permitted to access, download, copy, distribute, and use these materials in any way, even commercially, with proper attribution. For more information, see our copyright guidelines: http://journals.plos.org/plosone/s/licenses-and-copyright.

a. You may seek permission from the original copyright holder of Figure 1, 2, 6, 7, and 8 to publish the content specifically under the CC BY 4.0 license. 

Additional Editor Comments:

According to the Reviewers opinion it is necessary to emphasise the novelty of the research. The results presented in the paper should be compared with the results of other investigators. Additionally, other minor remarks are indicated in the reports should be considered.

Reviewers' comments:

Reviewer's Responses to Questions

**Comments to the Author**

1. Is the manuscript technically sound, and do the data support the conclusions?

Reviewer #1: Yes

Reviewer #2: Partly

2. Has the statistical analysis been performed appropriately and rigorously? 

Reviewer #1: Yes

Reviewer #2: Yes

3. Have the authors made all data underlying the findings in their manuscript fully available?

Reviewer #1: Yes

Reviewer #2: Yes

4. Is the manuscript presented in an intelligible fashion and written in standard English?

Reviewer #1: Yes

Reviewer #2: Yes

5. Review Comments to the Author

Reviewer #1: Dear Authors,

Thank you for your contribution to the field with this insightful study. However, I would like to request more information regarding the choice of the two specific loading scenarios - stair climbing and rising from a chair - employed in your experiments.

While these scenarios are undoubtedly relevant to the daily activities of patients undergoing revision total knee arthroplasty (TKA), the rationale behind selecting specifically these two scenarios over others is not clearly articulated in the manuscript. Understanding the reasoning for this selection is crucial, as it provides insight into the applicability and relevance of your findings to the broader context of post-operative patient activities.

Could you please elaborate on the following points:

The criteria or considerations that led to the selection of these particular scenarios.

Whether these scenarios are representative of the most common or most challenging activities encountered by patients post-revision TKA.

If there were any other scenarios considered during the planning phase of your study, and if so, why they were ultimately not included.

The selection criteria for the femurs used in your experiments, including any specific characteristics or parameters that were considered.

Clarifying these aspects would greatly enhance the understanding of the context and significance of your findings, thereby contributing to a more comprehensive interpretation of your research.

Reviewer #2: The presented problem is interesting, the experimental research was performed rigorously and described well.

However, the results regarding strains concentration and its character are rather obvious, and also expected by the Authors. Therefore, it is difficult to find novelty in the presented research.

Furthermore, the final conclusion that DIC method is “valuable method for future tests of biomechanical characteristics in different applications” is rather a general statement, as a lot of up-to-date biomechanical studies uses the DIC measurements already. Thus, application of the DIC measurements cannot be taken as the main novelty of the paper.

Finally, in my opinion, the authors did not find satisfactory biomechanical correlation between the end-of-stem pain and the obtained results. The presented research seems to be rather a pilot study and more numerical and clinical analysis should be added to make it comprehensive.

Additional remarks:

Fig. 4 is of poor quality, thus making the analysis of the results difficult.

How was the surface of the bone prepared before applying aerosol spray and the DIC measurements?

Reconsider, if the specimens with preexisting fracture should be included in the result section, as this fracture suggests that the stem was implanted improperly and therefore can affect the results.

6. PLOS authors have the option to publish the peer review history of their article (what does this mean?). If published, this will include your full peer review and any attached files.

Reviewer #1: No

Reviewer #2: No

---

## [Author Response · Author response to Decision Letter 0]

1 Mar 2024

Comments

 Response: We have updated the style of the manuscript and adapted the figure file naming and the style of the headings to PLOS ONE requirements.

 Response: We agree that ensuring adequate informed consent of the participants is important. In the ethical vote for this study this point is clearly addressed: Only specimens from so-called cadaver donors are used for this study. These individuals have kindly and voluntarily donated their bodies (to the Centre for Anatomy and Cell Biology - Medical University of Vienna) during their lifetime per written consent for the purposes of pre- and postgraduate teaching and scientific research. The anonymity of the study participants is strictly maintained and only age and gender are recorded. These two pseudonymised data are passed on to the project partner (B.Braun-Aesculap).

[I have read the journal's policy and the authors of this manuscript have the following competing interests:

Three of the authors (EMS, CS, TMG) are employees of Aesculap AG, a manufacturer of orthopedic implants.

RT and AG are paid consultants for Aesculap AG. AG is receiving royalties from 

Aesculap AG and is an unpaid consultant for DePuy Synthes. He is member of the Austrian Orthopaedic Society and of ”AE – Arbeitsgemeinschaft Endoprothetik”. RT is receiving royalties from Conformis and is a paid consultant for this company. RT has stock or stock options in OnPoint Surgical and receives support from Conformis and ZimmerBiomet as Principal Investigator. TMG is scientific member of the working group „Evaluations & Studies“ of the German National Joint Registry „Endoprothesenregister Deutschland“ (EPRD), Advisory Board Member of the EU Consortium SPINNER “Next generation of repair materials & techniques for spine surgery” and Chair of working group I “Introduction of Innovations” of the ”European Federation of National Associations of Orthopaedics and Traumatology” (EFORT) “Implant & Patient Safety Initiative”.]. 

 Response: You can find the revised conflict of interest statement including the confirmation of our adherece to all PLOS ONE data sharing policies on the previous page. 

 Response: Yes - all data are fully available without restriction.

All authors of this study agree to this. All relevant data are within the manuscript and its Supporting Information files.

5. Please include the reference section of your manuscript.

 Response: The reference section is included in the manuscript. 

6. We note that Figure 1, 2, 6, 7, and 8 in your submission contain copyrighted images. All PLOS content is published under the Creative Commons Attribution License (CC BY 4.0), which means that the manuscript, images, and Supporting Information files will be freely available online, and any third party is permitted to access, download, copy, distribute, and use these materials in any way, even commercially, with proper attribution. For more information, see our copyright guidelines: http://journals.plos.org/plosone/s/licenses-and-copyright.

 a. You may seek permission from the original copyright holder of Figure 1, 2, 6, 7, and 8 to publish the content specifically under the CC BY 4.0 license. 

 Response: Thank you for this input. All pictures included in this study were taken by the author, Dr. med. Elisabeth Sporer, and have not been published before. There are no third party figures. All authors agree with publication of the figures.

Additional Editor Comments:

According to the Reviewers opinion it is necessary to emphasise the novelty of the research. The results presented in the paper should be compared with the results of other investigators. Additionally, other minor remarks are indicated in the reports should be considered.

 Response: Thank you for your valuable input. We are happy that we were able to adress your recommendations and further improve the manuscript. We were able to compare our results to two previous studies (Kim et al. 2008, Completo et al 2012). The main novelty of our study is the experimental in-vitro design and the focus on the human femur. Completo et al. also used an in-vitro study design but analyzed tibiae. Our study is the first one showing strain concentrations at the tip of the femoral stem in revision total knee arthroplasty. We have tried to improve the manuscript by emphasizing the novely of the research.

Reviewers' comments:

Comments to the Author

Reviewer #1: Dear Authors,

Thank you for your contribution to the field with this insightful study. However, I would like to request more information regarding the choice of the two specific loading scenarios - stair climbing and rising from a chair - employed in your experiments.

While these scenarios are undoubtedly relevant to the daily activities of patients undergoing revision total knee arthroplasty (TKA), the rationale behind selecting specifically these two scenarios over others is not clearly articulated in the manuscript. Understanding the reasoning for this selection is crucial, as it provides insight into the applicability and relevance of your findings to the broader context of post-operative patient activities.

Could you please elaborate on the following points:

The criteria or considerations that led to the selection of these particular scenarios.

Whether these scenarios are representative of the most common or most challenging activities encountered by patients post-revision TKA.

If there were any other scenarios considered during the planning phase of your study, and if so, why they were ultimately not included.

 Response: Thank you for this insightful comment! We totally agree that the selection of adequate loading scenarios is crucial. Though the pain is described as activity related, there is no information in the literature about specific activities that provocate End-of-Stem Pain (Barrack et al., 1999, Barrack et al., 2004). Therefore, we have decided for two scenarios that are part of the daily activities of patients. We investigated one scenario with high bending moments at the tip of the stem (ChairRise90°) and one scenario with predominantly axial forces (Stairs20°). By choosing these loading scenarios we were able to analyze both biomechanical situations. We have added this information to the manuscript.

The selection criteria for the femurs used in your experiments, including any specific characteristics or parameters that were considered.

 Response: There were no specific inclusion criterias for the used donor bones except for no former surgery of the bone, no obviouse bone disease and no preexisting knee implantat.

Clarifying these aspects would greatly enhance the understanding of the context and significance of your findings, thereby contributing to a more comprehensive interpretation of your research.

Reviewer #2: The presented problem is interesting, the experimental research was performed rigorously and described well.

However, the results regarding strains concentration and its character are rather obvious, and also expected by the Authors. Therefore, it is difficult to find novelty in the presented research.

 Response: We agree that we expected to find strain concentrations at the tip of the stem. Nevertheless, there are no previous studies that were able to measure femoral strain concentrations. The finding has important implications for the future. It will allow for in-vitro development of new stem designs with a better biomechanical performance by using different materials or stem designs such as fluted stems, for example.

Furthermore, the final conclusion that DIC method is “valuable method for future tests of biomechanical characteristics in different applications” is rather a general statement, as a lot of up-to-date biomechanical studies uses the DIC measurements already. Thus, application of the DIC measurements cannot be taken as the main novelty of the paper.

 Response: We totally agree with the reviewer that DIC method has become an established method for strain measurement. Therefore, we have excluded the statement cited above from the manuscript.

Finally, in my opinion, the authors did not find satisfactory biomechanical correlation between the end-of-stem pain and the obtained results. The presented research seems to be rather a pilot study and more numerical and clinical analysis should be added to make it comprehensive.

 Response: Thank you for your feedback. We agree that there is more research necessary and that the correlation between End-of-Stem Pain and strain concentrations at the tip of the stem cannot be shown in our study due to the preclinical study design. Nevertheless, our study provides an important first step for the understanding of End-of-Stem Pain and builds the necessary basis for further research. Now that strain concentrations at the tip of the stem have been shown, future studies can focus on different materials and stem designs and their effect on surface strains.

Additional remarks:

Fig. 4 is of poor quality, thus making the analysis of the results difficult.

 Response: We have improved the resolution of Fig. 4.

How was the surface of the bone prepared before applying aerosol spray and the DIC measurements?

 Response: We approve of the reviewer´s suggestion of adding more information about bone preparation. Soft tissue was removed carefully using a bone curette, without damaging the surface of the bone. The bone was then cleaned using isopropanol before application of aerosol spray. We have added this information to the manuscript.

Reconsider, if the specimens with preexisting fracture should be included in the result section, as this fracture suggests that the stem was implanted improperly and therefore can affect the results.

 Response: We completely share the reviewer´s opinion that the preexisting fractures are important to keep in mind! We have discussed the reviewer´s comment intensely. On the one hand, preexisting fractures might influence the results. On the other hand, the implantation was performed following the recommended surgical technique and there was no reasonable suspicion for improper implantation. The small longitudinal fractures might also occure in some patients and may not be obvious on standard radiological imaging. We have therefore decided to include these specimens in the results section.

---

## [Decision Letter · Decision Letter 1]

27 Mar 2024

PONE-D-23-39275R1Strains on the human femur after revision total knee arthroplasty: An in vitro study using digital image correlationPLOS ONE

Dear Dr. Sporer,

Thank you for submitting your manuscript to PLOS ONE. After careful consideration, we feel that it has merit but does not fully meet PLOS ONE’s publication criteria as it currently stands. Therefore, we invite you to submit a revised version of the manuscript that addresses the points raised during the review process.

**ACADEMIC EDITOR: According to reviewers suggstions the paper still needs some minor chages. Please take them under consideration and prepare the final version of the paper.**

Please include the following items when submitting your revised manuscript:A rebuttal letter that responds to each point raised by the academic editor and reviewer(s). You should upload this letter as a separate file labeled 'Response to Reviewers'.A marked-up copy of your manuscript that highlights changes made to the original version. You should upload this as a separate file labeled 'Revised Manuscript with Track Changes'.An unmarked version of your revised paper without tracked changes. You should upload this as a separate file labeled 'Manuscript'.If applicable, we recommend that you deposit your laboratory protocols in protocols.io to enhance the reproducibility of your results. Protocols.io assigns your protocol its own identifier (DOI) so that it can be cited independently in the future. For instructions see: https://journals.plos.org/plosone/s/submission-guidelines#loc-laboratory-protocols. Additionally, PLOS ONE offers an option for publishing peer-reviewed Lab Protocol articles, which describe protocols hosted on protocols.io. Read more information on sharing protocols at https://plos.org/protocols?utm_medium=editorial-email&utm_source=authorletters&utm_campaign=protocols.

We look forward to receiving your revised manuscript.

Kind regards,

Pawel Klosowski, D.Sc.

Academic Editor

PLOS ONE

Journal Requirements:

Reviewers' comments:

Reviewer's Responses to Questions

**Comments to the Author**

1. If the authors have adequately addressed your comments raised in a previous round of review and you feel that this manuscript is now acceptable for publication, you may indicate that here to bypass the “Comments to the Author” section, enter your conflict of interest statement in the “Confidential to Editor” section, and submit your "Accept" recommendation.

Reviewer #1: All comments have been addressed

Reviewer #2: All comments have been addressed

2. Is the manuscript technically sound, and do the data support the conclusions?

Reviewer #1: Partly

Reviewer #2: Yes

3. Has the statistical analysis been performed appropriately and rigorously? 

Reviewer #1: I Don't Know

Reviewer #2: Yes

4. Have the authors made all data underlying the findings in their manuscript fully available?

Reviewer #1: Yes

Reviewer #2: Yes

5. Is the manuscript presented in an intelligible fashion and written in standard English?

Reviewer #1: Yes

Reviewer #2: Yes

6. Review Comments to the Author

Reviewer #1: Thank you for your contribution to the field with this insightful study.

I have been reviewing your manuscript and find it to be quite promising. However, there are a couple of areas where I believe further clarification and elaboration would greatly enhance the quality and impact of your work.

Data Analysis Methodology: Regarding the method of measuring surface deformation using Digital Image Correlation (DIC), I would appreciate a more detailed technical explanation and a thorough discussion of its efficacy. Specifically, we need more information on the accuracy and reproducibility of measurements achieved through DIC, as well as the advantages of this method compared to existing techniques.

Analysis of Fracture Patterns: While your manuscript presents results on fracture patterns post-excessive loading, it would greatly benefit from a more explicit explanation of how these patterns contribute to the understanding of the biomechanical mechanisms of End-of-Stem Pain. Additionally, further information is needed on quantitative approaches to fracture pattern analysis and how these results should be interpreted clinically.

Reviewer #2: DearAuthors,

thank you very much for your effort in preparing responses and corrections.  

I am still interested if there are any other studies regarding strains on the bone and their correlation with any pain. Do you know what level of strain results in pain? Was it any time before tested or checked? Maybe on the other bones or tissues? In my opinion, a discussion on this point is worth to be included in the paper, which could make your research more comprehensive. If such details are not available, they should be strongly highlighted or explained in the text. 

In the discussion section, you mentioned about similarities between your results of the obtained strain levels with the outcomes of other researchers. It would enhance the discussion if you put more detailed information about the comparison of these results, e.g. - what was the highest strain level in your research versus the highest one obtained by others? what was the range of strains obtained by you and the others? does this range change due to the type of bone analyzed or the method of measurement applied? More numbers will make the discussion more clear and informative. In my opinion, statements, that “the results are similar” is too general.

7. PLOS authors have the option to publish the peer review history of their article (what does this mean?). If published, this will include your full peer review and any attached files.

Reviewer #1: No

Reviewer #2: No

---

## [Author Response · Author response to Decision Letter 1]

10 May 2024

Reviewers´ commernts:

Data Analysis Methodology: Regarding the method of measuring surface deformation using Digital Image Correlation (DIC), I would appreciate a more detailed technical explanation and a thorough discussion of its efficacy. Specifically, we need more information on the accuracy and reproducibility of measurements achieved through DIC, as well as the advantages of this method compared to existing techniques.

Strain gauges have been considered the gold standard for strain measurement for a long time, but they only offer strain measurement at discrete points. DIC is a non-contact method to measure strains and it allows for a full-field strain analysis. In a study by Correa et al. (Correa et al. 2018), correlation between strain measurement using strain gauges and DIC has been shown. With the system used in our study an approximated accuracy for strain of 0.1% was achieved for both in-plane and out-of-plane measurement. The reproducibility is approximately +/- 0.2%. We have added this information to the manuscript.

Analysis of Fracture Patterns: While your manuscript presents results on fracture patterns post-excessive loading, it would greatly benefit from a more explicit explanation of how these patterns contribute to the understanding of the biomechanical mechanisms of End-of-Stem Pain. Additionally, further information is needed on quantitative approaches to fracture pattern analysis and how these results should be interpreted clinically.

Until now, the amount of biomechanical influence of the stem in knee arthroplasty has still been unknown. The specific fracture pattern found in this study suggests that there is a high biomechanical impact of the stem. It is also still unclear why only certain patients suffer from End-of-Stem Pain whereas others do not experience End-of-Stem Pain. In our study, only in some specimens overcritical load led to the stem pattern fracture. This may suggest that the biomechanical influence of the stem is higher in some patients than in others. This may explain why only some patients suffer from End-of-Stem Pain and underlines the importance of individualized selection of the right implant. We have added this to the discussion of our manuscript.

I am still interested if there are any other studies regarding strains on the bone and their correlation with any pain. Do you know what level of strain results in pain? Was it any time before tested or checked? Maybe on the other bones or tissues? In my opinion, a discussion on this point is worth to be included in the paper, which could make your research more comprehensive. If such details are not available, they should be strongly highlighted or explained in the text.

Unfortunately, to our knowledge there are no studies that report which level of strain results in pain, neither at the human femur nor at other bones. We strongly agree that further studies are required in this field.

In the discussion section, you mentioned about similarities between your results of the obtained strain levels with the outcomes of other researchers. It would enhance the discussion if you put more detailed information about the comparison of these results, e.g. - what was the highest strain level in your research versus the highest one obtained by others? what was the range of strains obtained by you and the others? does this range change due to the type of bone analyzed or the method of measurement applied? More numbers will make the discussion more clear and informative. In my opinion, statements, that “the results are similar” is too general.

Thank you for this feedback, we agree that discrete numbers help to interpret the results. We have added a paragraph precisely comparing the results of both studies to our

---

## [Decision Letter · Decision Letter 2]

29 May 2024

Strains on the human femur after revision total knee arthroplasty: An in vitro study using digital image correlation

PONE-D-23-39275R2

Dear Dr. Sporer,

We’re pleased to inform you that your manuscript has been judged scientifically suitable for publication and will be formally accepted for publication once it meets all outstanding technical requirements.

Kind regards,

Pawel Klosowski, D.Sc.

Academic Editor

PLOS ONE

Additional Editor Comments (optional):

Reviewers' comments:

Reviewer's Responses to Questions

**Comments to the Author**

1. If the authors have adequately addressed your comments raised in a previous round of review and you feel that this manuscript is now acceptable for publication, you may indicate that here to bypass the “Comments to the Author” section, enter your conflict of interest statement in the “Confidential to Editor” section, and submit your "Accept" recommendation.

Reviewer #1: All comments have been addressed

Reviewer #2: All comments have been addressed

2. Is the manuscript technically sound, and do the data support the conclusions?

Reviewer #1: Yes

Reviewer #2: Yes

3. Has the statistical analysis been performed appropriately and rigorously? 

Reviewer #1: Yes

Reviewer #2: Yes

4. Have the authors made all data underlying the findings in their manuscript fully available?

Reviewer #1: Yes

Reviewer #2: Yes

5. Is the manuscript presented in an intelligible fashion and written in standard English?

Reviewer #1: Yes

Reviewer #2: Yes

6. Review Comments to the Author

Reviewer #1: Thank you for addressing my previous comments.

I am now satisfied with the revisions and accept the manuscript.

Reviewer #2: (No Response)

7. PLOS authors have the option to publish the peer review history of their article (what does this mean?). If published, this will include your full peer review and any attached files.

Reviewer #1: No

Reviewer #2: No

---

## [Editor Report · Acceptance letter]

3 Jun 2024

PONE-D-23-39275R2 

PLOS ONE

Dear Dr. Sporer, 

I'm pleased to inform you that your manuscript has been deemed suitable for publication in PLOS ONE. Congratulations! Your manuscript is now being handed over to our production team.

Kind regards, 

on behalf of

Prof. Pawel Klosowski 

Academic Editor

PLOS ONE